# Evaluating the Nutritional Properties of Food: A Scoping Review

**DOI:** 10.3390/nu14112352

**Published:** 2022-06-05

**Authors:** Pei Wang, Jiazhang Huang, Junmao Sun, Rui Liu, Tong Jiang, Guiju Sun

**Affiliations:** 1Key Laboratory of Environmental Medicine and Engineering of Ministry of Education, Department of Nutrition and Food Hygiene, School of Public Health, Southeast University, Nanjing 210009, China; wangpei929@seu.edu.cn; 2Institute of Food and Nutrition Development, Ministry of Agriculture and Rural Affairs, Beijing 100081, China; huangjiazhang@caas.cn (J.H.); sunjunmao@caas.cn (J.S.); liurui@caas.cn (R.L.); 3Beijing Billion Power Nutrition Technology Co., Ltd., Beijing 100069, China; jiangtong@df2000.cn

**Keywords:** nutrition evaluation, nutritional value of food, evaluation method, evaluation indicator

## Abstract

There are many methods or indicators used for evaluating the nutritional value of foods; however, it is difficult to accurately reflect the comprehensive nutritional value of a food with a single indicator, and a systematic evaluation system is lacking. In this article, we systematically summarize the common evaluation methods and indicators of the nutritional value of foods. The purpose of this review was to establish an evaluation procedure for nutritional properties of foodstuffs and to help scientists choose more direct and economical evaluation methods according to food types or relevant indicators. The procedure involves the selection of a three-level evaluation method that covers the whole spectrum of a food’s nutritional characteristics. It is applicable to scientific research in the fields of agricultural science, food science, nutrition, and so on.

## 1. Introduction

A food is a complex combination of nutrients and other compounds that act synergistically within the food and across food combinations [1]. Food plays an integral role in human metabolism, digestion, and growth. Disciplines related to food range widely, including agricultural science, food science, nutrition, and so on. The role of agriculture in reducing undernutrition is widely recognized [2]. Nutrition-sensitive agriculture [3], a new agricultural model, encourages agricultural researchers to pay more attention to the food’s nutritional quality. Nutrition-sensitive agriculture programs could help to scale up nutrition-specific interventions [4]. Agricultural economists have put forward the idea of identifying foods with good nutritional qualities and using this to set their prices [5,6]. In the field of food science, scientists are more concerned about processing technologies, quality characteristics, packing, storage and preservation, and nutrient content [7,8,9,10,11,12]. Processed foods have a negative image among consumers and experts regarding their food–health imbalance [13]. It is still controversial whether the dietary contribution of ultraprocessed foods (UPFs) largely determines the overall nutritional quality of contemporary diets [14]. Nutrition science is mainly concerned with the effects of nutrients on human health, and simplifies complex nutritional requirements into manageable recommendations in the form of dietary guidance for the purpose of avoiding diseases [15].

In traditional nutrition, the nutritional value of food is reflected by the type and quantity of nutrients. After nearly a century of development, increasing numbers of studies have shown that the majority of chronic diseases are caused by nutrient deficiencies, such as insufficient intake of a single nutrient (e.g., vitamins or minerals) or unbalanced dietary patterns. Dietary components are consumed in combination and are correlated with one another [16]. Nutrition nowadays is not limited to the supply of nutrients to human beings, but is also an opportunity to prevent the onset of disease and maintain health. As an alternative to adequate nutrition, the concept of optimal nutrition has been presented, with the goal of improving well-being and mitigating the harmful health consequences by helping to prevent or control most chronic diseases, aid in the regulation of sleep and mood, and prevent fatigue [17]. Therefore, an important direction of nutrition development in the future is to meet the optimal nutritional needs of different groups through to the nutritional properties of foodstuffs. To evaluate the nutritional value of a food, we should also consider the proportion of nutrients and the degree of human digestion, absorption, and utilization. The true bioavailability of a nutrient is intrinsically coupled to the specific food matrix in which it occurs, but this remains poorly considered in nutrition science [18]. Consumers find it hard to distinguish what is truly good nutritious food [19]. Different experts have used different perspectives and methods to evaluate the nutritional value of foods. In addition, some individuals on social media, lacking professional knowledge, may exaggerate or belittle the value of food as a publicity gimmick. Rabassa [20] described the methodology, characteristics, and contents of Nutri-media, a web-based resource, to evaluate the veracity of nutrition claims disseminated to the public by the media. However, it is still not easy for the public to make informed choices about the nutritional value of foods.

As an interdisciplinary research field, the evaluation of nutritional value of food is particularly important and popular. It can not only be applied in scientific research, but can also be used to guide the formulation of nutrition policies and standards and consumer-oriented knowledge popularization. Therefore, it is necessary to carry out nutrition evaluation studies through valid and effective evaluation systems and consideration of multidimensional indicators.

The current problems are as follows. “Nutrition evaluation” is a commonly used concept in clinical settings, and the evaluation object is the human body. Research on evaluation of foods’ nutritional characteristics varies greatly, and there are differences in the understanding and definition of the concept in different disciplines. For example, food scientists usually analyze the nutritional value of a food according to the type and content of nutrients in the food [21,22,23,24,25,26]. However, the type and content of nutrients may not be enough to reflect the nutritional value of the food. Furthermore, nutrition practitioners and researchers emphasize the final effects of the nutrients or the food system on human health [27,28,29,30,31,32], and thus the nutritional value of a food should be evaluated through another systematic evaluation scheme. Therefore, it is necessary to distinguish the different methods used for evaluation of the nutritional value of food.

We designed an evaluation procedure framework in order to provide researchers with ideas for nutrition evaluation and to promote the application and development of nutrition evaluation of foodstuffs.

## 2. Materials and Methods

A literature search was conducted for all articles indexed by Web of Science, PubMed, and Scopus up to February 2022. The search strategies were completed using keywords including “food”, “food quality”, “diet, food and nutrition”, “evaluation”, “evaluation method”, “evaluation tool”, “evaluation indicator”, “nutrients” and “method”. Keywords were amended slightly for each database. The full list of search terms is given in Appendix A.

Through the established search strategy, 3386 studies in total were initially assessed, of which 124 duplications and 3115 irrelevant studies were excluded after reviewing the title and abstracts, leading to a total of 124 articles for further assessment. After reading the full text of the remaining articles, 109 articles were removed for the following reasons: (ⅰ) no proper evaluation indicators (*n* = 69); (ⅱ) no relevant outcomes (*n* = 19); (ⅲ) inappropriate types of articles, such as meeting abstracts or patents (*n* = 21). Finally, 16 studies (including one study added from the references) were eligible for the present quantitative synthesis. Two researchers carried out independent eligibility screening using Endnote, and disputes were resolved via discussion with a third researcher. A detailed flow diagram of the selection process is shown in Figure 1.

Studies were selected if they met the following criteria: (ⅰ) the research subject must be food or components of food; (ⅱ) the study was performed on foodstuffs or included human trials; (iii) the indicators reflected the food’s quality; (ⅳ) the study reported data with the definition and components of the evaluation methods explained. The exclusion criteria were the following: (ⅰ) animal or cell experiments; (ii) studies without proper indices to reflect the food’s qualities, such as human health status indicators; (ⅳ) invalid information such as patents; and (v) articles that were not in English.

## 3. Common Methods for Evaluating the Nutritional Value of Food

According to the classified catalogue of food production licenses revised by China’s State Administration for Market Regulation in 2020, foods can be divided into 32 categories and more than 100 subcategories. The classifications of food are illustrated as a sunburst chart (Figure 2).

Unlike human nutrition assessments, evaluating the nutritional value of a food refers to scientific judgments of the nutritional value of the food according to valid scoring tools and evaluation standards [33].

Effective scientific evaluation methods for the nutritional value of food are based on the selection of valid indices and criteria. A range of evaluation methods have been established and are widely used, as shown in Table 1. The first aim of this study was to collect all the methods that have been used to date and to display these systematically, and then to analyze these in depth to compare the methods and provide new theoretical guidance or stimulate new methods for evaluating the nutritional value of food. Hence, we considered the differences in the evaluation methods, especially their strengths and weakness, as shown in Table 2. According to this evaluation approach, the evaluation methods were divided into categories as follows.

### 3.1. Evaluation Methods Based on Food Nutrients

#### 3.1.1. Nutrient Comparison Method

It is simple and effective to evaluate nutritional value via the types and quantities of nutrients in food [46]. After being detected, analyzed, and calculated, the nutrients of certain foods are compared with the values in the food composition table (FCT). Next, by using the recommended intake given by the dietary reference intakes (DRIs), we can determine the preliminary nutritional value of the food [47,48,49]. Nutrient profiling, which is the technique of rating or classifying foods on the basis of their nutritional value [50], is a quick way to regulate nutrition labels, health claims, and marketing and advertising [51].

Food composition tables and databases (FCT/FCDB) collate data on the energy and nutrient contents of foods for a certain country or region. Around 100 countries or regions have published at least one FCT/FCDB, although many of them are outdated and vary considerably in terms of data quality, documentation, and accessibility [52]. Numerous FCDBs have sprung up since the 1980s, including Latinfoods, Asiafoods, Oceaniafoods, Norfoods, and Eurofoods. Eurofoods includes both Eastern and Western Europe [53]. American food nutrition data are provided by FoodData Central (FDC), the center of the US Department of Agriculture’s (USDA’s) food composition information website. The FDC provides five different data types, namely foundation foods (FF), experimental foods (EF), standard reference legacies (SR Legacy), the Food and Nutrient Database for Dietary Studies (FNDDS), and the Global Branded Foods Products Database (GBFPD). The FDC provides reliable, web-based, transparent, and easily accessible information about the nutrients and other components of foods to meet the increasingly diverse needs of many audiences [54]. Food composition data (FCD) with up to 87 core components for approximately 600 foods have been added to the New Zealand Food Composition Database (NZFCDB) since 2010 [55]. China’s FCT [56] aims to help consumers understand the nutritional content of food and to make informed food choices, including general nutrition data on more than 4710 raw plant and animal materials. In addition, we noted that food composition tables are rarely consistent across countries. Many foods are defined or presented in different ways, making comparison of nutrient compositions difficult [53].

#### 3.1.2. Chemical Scoring Method

Protein, a class of organic macromolecule, is the basic organic matter of cells and is the main agent in the activities of living. There would be no life without protein. Amino acids are the basic components of proteins. Hence, protein quality has long been used to reflect the nutritional value of food. Chemical scoring involves the establishment of a corresponding mathematical model according to the nutritional composition, and is mainly used for the evaluation of protein.

The amino acid score (AAS) [34] is a score used to evaluate the proximity of an essential amino acid in the protein to be tested to the corresponding essential amino acid in the reference protein model. It is a widely used index for evaluating the nutritional value of food protein. In 1990, the Food and Agriculture Organization of the United Nations (FAO) developed an approach for quantifying protein quality, which is called the protein-digestibility-corrected amino acid score (PDCAAS) [35]. A PDCAAS of 1.0 means that all of the minimal requirements for indispensable amino acid (IAA) intake would be met if the amount of the test protein eaten were equivalent to the estimated average requirement (EAR) for protein [36]. However, the truncation of the PDCAAS is 1.0, which means PDCAAS cannot distinguish the relative quality of high-quality dietary proteins. Therefore, Wolfe et al. [36] and Mahtai et al. [57] argued that digestible amino acid scores (DIAAS) can describe the protein quality of food more accurately than the PDCAAS. In contrast to the PDCAAS, the DIAAS is not truncated for a single-source protein, thereby theoretically enabling the ranking of all dietary proteins by their quality. The essential amino acid index (EAAI) [37], another indicator, is a geometric mean used to make an overall comparison between all essential amino acids in the protein to be tested and all essential amino acids in the reference protein model.

Although protein plays an irreplaceable role in the nutritional value of food, it does not represent the overall nutritional characteristics of a food. Some indices of the nutritional value of whole foods have been developed and applied. Some indexes that attempt to quantify the nutrient density of foods have been developed, such as calorie-to-nutrient scores, nutrients-per-calorie indexes, and nutrient-to-nutrient ratios [40].

The food nutritional quality index (the index of nutrition quality, INQ) [38] is a method of quantitatively and qualitatively analyzing single foods, meals, and diets, which has special significance in assessing clinical nutritional issues [58]. The naturally nutrient-rich (NNR) score [40], which is based on mean percentage daily values (DVs) for 14 nutrients in 2000 kcal of food, can be used to assign nutrient density values to foods within and across food groups. The nutrient density approach can be a valuable tool for nutrition education and dietary guidance. The calories-for-nutrients (CFN) [40] score is defined as the cost in calories that is required to obtain an additional 1% DV for a range of key nutrients, and thus it can be used to directly assess the relationship between food energy and nutrient value.

### 3.2. Evaluation Methods Based on Food Function

A growing number of studies have shown that a single or limited nutrient assessment cannot objectively reflect the nutritional value of a particular food. A food matrix may exhibit a different relationship with health indicators from individual nutrients studied in isolation [59]. Therefore, diet and food group quality have been proposed as indicators of nutrition from the perspective of balance, variety, adequacy, and moderation. From the perspective of epidemiology, chemical components are only the nutritional basis of food, and the nutritional value should also reflect its beneficial function in human body, such as the treatment and improvement of cardiovascular diseases, diabetes, and other chronic diseases.

On the basis of US dietary recommendations from Diet and Health, Patterson et al. [42] proposed an index for the evaluation of diet quality (DQI) in 1994. Initially, the DQI consisted of eight components, and the values ranged between 0 (excellent diet) and 16 (poor diet). During 1999, in order to incorporate improved methods of estimating food servings and to develop and incorporate measures of dietary variety and moderation, Haines et al. [43] revised the DQI and created the DQI–Revised (DQI-R). This new index consisted of 10 components, and the score ranged from 0 and 100. However, in 2003, in order to evaluate healthiness of diets not only within a country for monitoring purposes but also across countries for comparative work, Kim et al. [44] revised the index again to create the DQI–International (DQI-I). The DQI-I emphasizes the scores of four diet categories, which are variety (overall variety and variety within protein sources), adequacy (fruits, vegetables, grains, fiber, protein, iron, calcium and vitamin C), moderation (total fat, saturated fat, cholesterol, sodium, empty-calorie foods), and overall balance (macronutrient ratio, fatty acid ratio).

The glycemic index (GI) [45] refers to the glycemic effect of the available carbohydrates in a food relative to the effect of an equal amount of glucose. The clinical application of GI is to guide people to choose appropriate foods according to their blood glucose levels, especially those with diabetes. The boundary scores of the GI are 55 and 70. Generally speaking, foods with GI > 70 are foods with a high glycemic index. High-GI food has a high absorption rate after entering the gastrointestinal tract; thus, the glucose is released rapidly, resulting in high peak in blood glucose. In contrast to high-GI foods, foods with GI < 55 have a lower food glycemic index and can stay in the gastrointestinal tract for a long time, with a low absorption rate, a slow release of glucose, a low peak value, and a slow rate of decline. Therefore, organizing a suitable diet using the GI is of great benefit for regulating and controlling human blood glucose.

### 3.3. Evaluation Methods Based on Sensory Perception

In the food cultures of some countries, the “healthiness” of a food is always associated with “tastiness”. Although some researchers have argued that people usually hold the perception that “healthy equals less tasty” [60], others have insisted that this perception could be diminished [61] in some way or that “healthy = tasty” in different regions [62,63]. For example, in French epicurean culture or Chinese food culture, in which food is valued primarily for its tastiness, it is possible that the perception of an inverse relationship between healthiness and tastiness is held with less conviction [60]. If the taste of a nutritious food is not accepted by consumers, its nutritional value is meaningless [64,65]. Hence, sensory attributes are non-negligible and “taste” should be included in a food evaluation system [66].

Sensory evaluations combined with dietary suggestions can be summarized as a new method for evaluation of food’s nutritional value. Traditional sensory evaluation is mainly used in the development and marketing of new foods to satisfy consumer tastes [67,68]. Reliable sensory analysis requires a well-trained manual evaluation team. The sensory test protocol contains a set of techniques for accurately measuring human responses to food [69]. For example, the flavor wheel, a sensory evaluation tool, has been widely used for coffee, wine, tea, fermented food, and so on [70,71,72,73], and also serves as a communication tool among all components of the industry, including tasters, plants, retailers, exporters and importers, producers, baristas, and consumers [74]. Consumers can accurately obtain the sensory information of a particular tested food. One example of the flavor wheel system is the Panda Guide, a list of Chinese high-quality agricultural products launched by Sinochem Agriculture, which shows evaluation indexes related to agricultural products according to the appearance, flavor, taste, texture, and so on. This provides a standardized evaluation method of agricultural products combined with instrument detection data. Furthermore, nutrition information could be added to the flavor wheel, which would be conducive to the popularization of nutrition and health knowledge.

## 4. Construction of a Procedure for Evaluating the Nutritional Value of Food

### 4.1. Steps in Evaluating the Nutritional Value of Food

The purpose of breaking the evaluation procedure into levels is to help researchers clarify the position of the evaluation. Referring to the common food types and evaluation purposes, we have indicated the evaluation steps for the rational selection of appropriate methods, as shown in Figure 3.

Differences exist in food-related disciplines, so it is easy to see that an appropriate evaluation method can be chosen according to our purpose. For example, food scientists may be more concerned about nutrient losses during food processing, and the application of a Level 1 evaluation is sufficient to reflect the nutritional value of certain foods. Meanwhile, in the field of preventive medicine, scientists are committed to studying the roles of food and nutrients in the prevention and treatment of chronic diseases. Therefore, a Level 2 evaluation is necessary to verify or explore the health effects of food. In addition, with a valid evaluation scheme, the results are also conducive to popularizing knowledge on nutrition and helping consumers to choose food.

### 4.2. Levels of Evaluation of Nutritional Value of Food

According to the methods for evaluating the nutritional value of food summarized above, it is obvious that the food category, evaluation purpose, and research depth directly affect the selection of an appropriate evaluation method. Hence, for the purpose of achieving a valid and adequate evaluation, we classified the nutrition evaluation procedures as shown in Figure 4.

Level 1 evaluations involve nutrient detection and analysis. In other words, they detect and test the nutrients of food through physical and chemical analysis methods, and compare them with the nutrients in a reference food or FCT to interpret the food’s nutritional value. Level 1 evaluation methods can be used for the majority of nutrition evaluations and for descriptions of the nutritional characteristics of food, and thus can be applied to the evaluation of agricultural products, prepackaged food, etc.

On the basis of a Level 1 evaluation, Level 2 evaluation methods emphasize the health effects of a food and its components from a clinical perspective. The biological nutritional value of food can be reflected by various nutritional indexes, and the health effects can be further explored and verified by animal studies or human trials. Level 2 evaluation is applicable for verifying the beneficial effects of food and exploring its functional components.

Level 3 evaluations explore the mechanisms of the effects of nutrients or food groups on human health, beyond Level 2 evaluations. The difference between a Level 3 evaluation and the abovementioned evaluation methods lies in their support of clinical nutrition, molecular nutritional biology, and other disciplines to study the physiological functions of nutrients in the body at the gene and cell level, and to clarify the nutritional physiology and mechanisms of action.

## 5. Discussion

### 5.1. Prospects for Evaluating Food Nutrition

In the future, methods for evaluating food nutrition should be developed further. More nutrient scoring systems and overall food group evaluating systems should be developed, especially in the field of evaluating the nutrition of dietary patterns.

In addition, nutrition evaluation methods should be widely applied in the food industry. First, evaluations can assist in developing new products. Products with beneficial effects can be targeted to consumer groups more accurately. For example, research into low-GI foods provides guidance for members of the population with an abnormal blood glucose metabolism. The product’s ingredients can be determined through the GI by selecting appropriate raw materials, optimizing the processing parameters, and adjusting the processing methods.

Secondly, it can also provide scientific evidence for the formulation of nutritional policy. Evaluations of foods’ nutritional value can provide medical evidence through well-designed animal experiments or human trials, to help governments formulate relevant policies and guidelines. For example, evaluations of nutrient intake can guide the revision of DRIs.

Moreover, with an improvement in citizens’ health awareness, the demand for nutritious and healthy food choices has gradually become stronger. Research on evaluating the nutritional value of food can promote the development and improvement of food labeling and certification systems, and thus help consumers to choose healthy foods. In recent years, government agencies, nonprofit organizations, and food enterprises in the United States, Britain, South Korea, Thailand, Malaysia, Singapore, Australia, New Zealand, Canada, Germany, the Netherlands, Belgium, and other countries have designed icons to summarize the main nutritional information and characteristics of food [75]. These icons and the nutrition evaluation system used to judge whether food can be labeled with icons are called front-of-package (FOP) systems. The establishment of an FOP system is based on an accurate evaluation of the nutrient content of foods. China also launched its domestic FOP standard and the “healthy choice” logo at the end of 2017. At present, many certification bodies in China can certify products as “healthy food” and “nutritious food”, and consumers can choose food with the guidance of these labels.

Food is a complex combination of nutrients and other compounds, and different groups have various needs in terms of foodstuffs. In the future, food scientists could use the experience of precision medicine to develop precision food, with accurate evaluation of the nutrients, taste, and function to meet individuals’ needs for optimal nutrition.

Additionally, with the world paying attention to environmental protection, the food industry should develop in the direction of sustainability, lower carbon emissions, and less pollution in the future. The evaluation indicators of foodstuffs may not only be limited to the nutritional content or the physiological effects on human body, but may also include indices of the impact on the environment in the process of planting, growing, and processing. Therefore, future evaluations of the nutritional value of food should be part of a diversified and comprehensive system.

### 5.2. Existing Problems and Improvements

With the development of agricultural science, food science, and nutrition, research hotspots in various fields continue to cross over and enter different fields. Research into evaluating the nutritional value of food plays a significant role in this. At present, China’s agricultural model has gradually entered the stage of nutrition sensitivity and green development, agricultural science and technology continue to break through the bottlenecks, and the nutritional standards of agricultural products are constantly being established. The cultivation, planting, and breeding of products with high nutritional value are related to research into nutritional evaluation. Achievements in food science have accelerated the vigorous development of the food industry. Through upgrades in processing technology, packaging materials, and production equipment, the form of food also keeps changing. The food industry has shifted from its traditional focus on taste to considerations of nutritional value, and from the traditional detection of health and safety indicators to the detection of nutrients and special efficacy components. In the field of preventive medicine, the relationship between nutrients and chronic diseases is still the focus of current research. Dietary patterns and balanced nutrition are inseparable from studies evaluating the nutritional value of food.

There are many disciplines related to food, including agricultural science, food science, nutrition, economics, and so on. The nutritional properties of foodstuffs are one of the foundations of those disciplines, so reflecting them objectively and accurately is particularly important. In various disciplines, the approaches taken to evaluation are not the same, mainly due to the different research depths, academic levels, and purposes of the researchers. Some researchers believe that the nutrient content of a food itself can directly represent its nutritional value. The more types of nutrients and the higher their contents, the more nutritious a food must be. In fact, due to the impact of processing technology, storage conditions, and nutrient absorption and utilization rates, the nutrient content of a food itself may not directly reflect its nutritional value. In addition, the development of health foods and formula foods for special medical purposes also requires valid evaluation methods. The form and dissemination of evaluation results about the nutritional value of food should not only meet the requirements of relevant laws and regulations, but also facilitate consumers’ understanding. Therefore, it is worth establishing a universal and logical procedure to guide scientists’ research on evaluation of the nutritional value of food.

In the field of research on evaluation of the nutritional value of food, we can learn from nutrition assessment in medical science, which is the evaluation and measurement of nutritional variables for assessing the level of nutrition or the nutritional status of the individual. The object of our research is the foodstuff itself. We hope that food scientists can be inspired by this work to establish food-related evaluation scales, and to select valid, objective, and accurate nutrition evaluation methods according to their research purposes and funding conditions, the characteristics of the food itself, and other factors.

## 6. Conclusions

In this article, we summarized the common indices and evaluation methods used to reflect the nutritional characteristics of food. Furthermore, we formulated new theoretical guidance by designing a framework for a procedure of evaluating the nutritional value of food. The procedure involves the selection of a three-level evaluation method that covers the whole spectrum of food’s nutritional characteristics. It is applicable to scientific research in the fields of agricultural science, food science, nutrition, and so on. With the guidance of this procedure, scientists can choose valid methods to fairly reflect the nutritive value of foodstuffs according to their research purposes. Moreover, the results obtained by using this procedure can provide consumers with comprehensive and systematic food knowledge.

## Figures and Tables

**Figure 1 nutrients-14-02352-f001:**
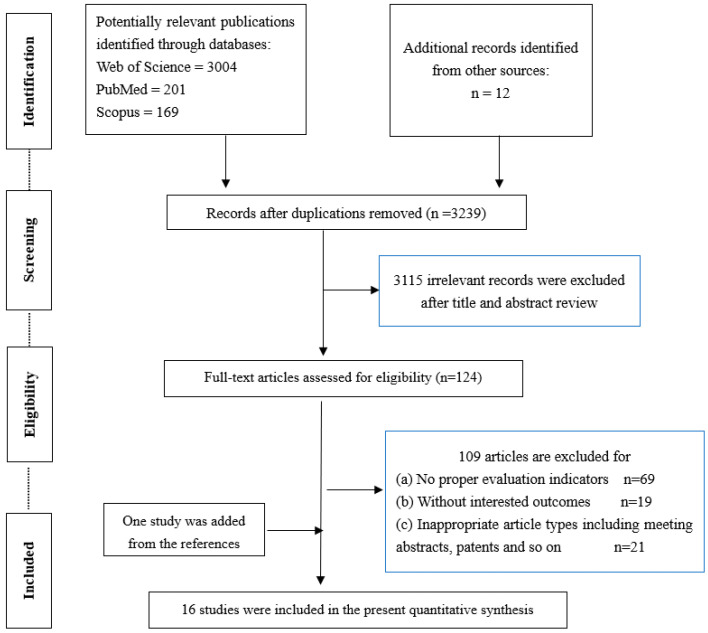
The flow diagram for article selection.

**Figure 2 nutrients-14-02352-f002:**
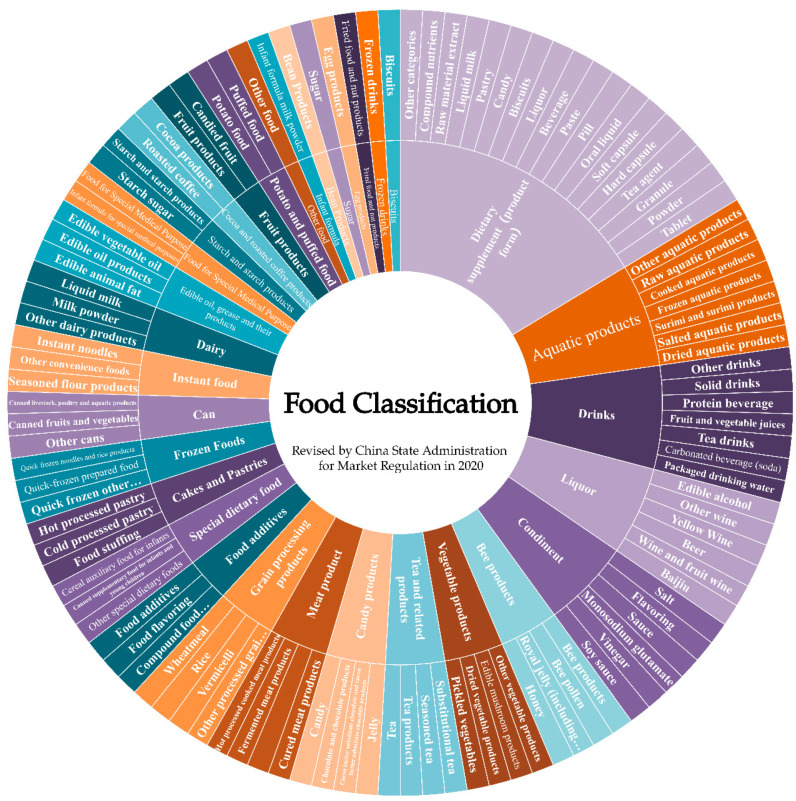
Sunburst chart of food classifications (The classified catalogue of food production licenses was formulated by China’s State Administration for Market Regulation. An application for a food production license in China is made according to the following food categories: food processing products, edible oil, oil and its products, condiments, meat products, dairy products, beverages, convenience foods, biscuits, canned food, frozen drinks, quick-frozen foods, potatoes and puffed foods, candy products, tea and related products, alcohol, vegetable products, fruit products, fried food and nut products, egg products, cocoa and roasted coffee products, sugar, seafood products, starch and starch products, pastries, bean products, bee products, health foods, formula food for special medical purposes, infant formula food, special dietary food, and other foods.) (from China’s State Administration for Market Regulation in 2020).

**Figure 3 nutrients-14-02352-f003:**
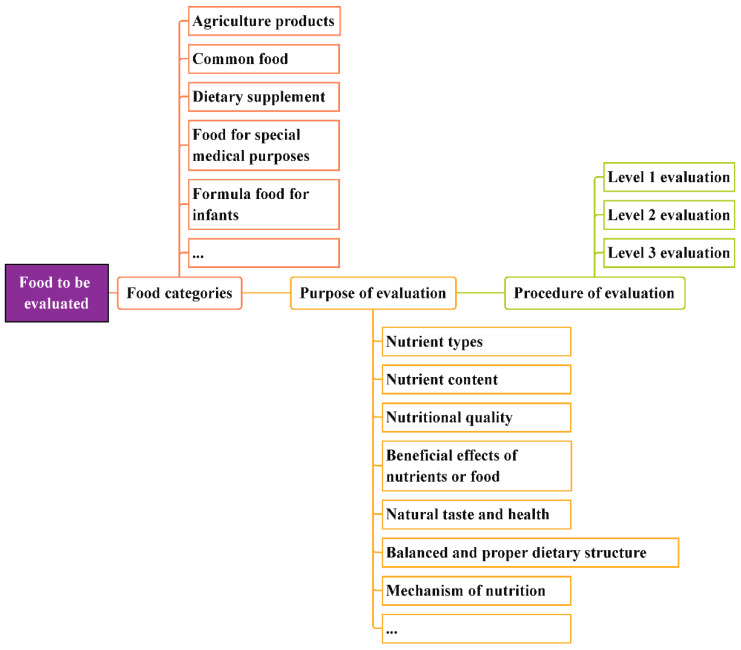
Procedures for evaluating the nutritional value of food.

**Figure 4 nutrients-14-02352-f004:**
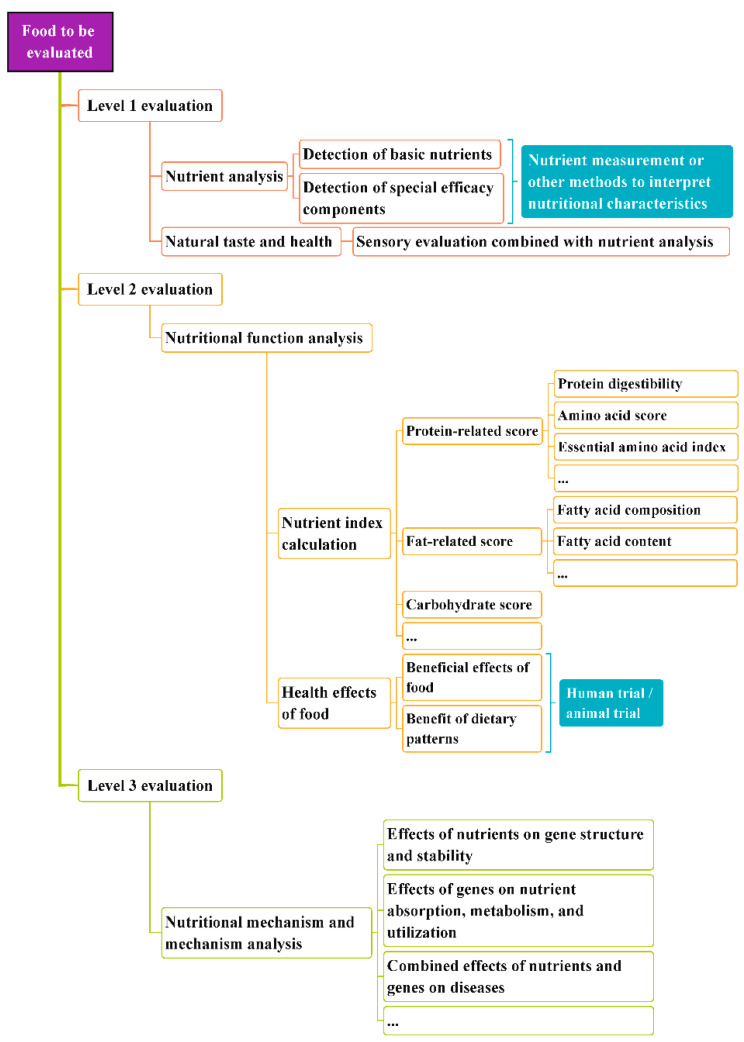
Levels of nutritional value evaluation procedures.

**Table 1 nutrients-14-02352-t001:** Common methods for evaluating the nutritional value of food.

Evaluation Methods	Author	Year	Definition	Program Evaluated	Computing Method	Components	Evaluation of Quality Index
Amino acid score (AAS)	Bano et al. [34]	1982	A score used to evaluate the proximity of an essential amino acid in the protein to be tested to the corresponding essential amino acid in the reference protein model.	Protein quality	AAS = amino acid content in sample protein (mg/g)/corresponding essential amino acid content in the FAO or WHO scoring standard model (mg/g)	Amino acid content in a sample of protein	Amino acid score
Protein-digestibility-corrected amino acid score (PDCAAS)	Eggum et al. [35]	1991	Based on the ratio of the amount of the first-limiting dietary indispensable amino acid in the protein source to the amino acid requirements of a 1–2-year-old child corrected for protein digestibility based on true fecal nitrogen digestibility and using a growing rat as a model for the adult human.	Protein quality			
Digestible indispensable amino acid score (DIAAS)	Wolfe et al. [36]	2016	Based on the relative digestible content of indispensable amino acids (IAAs) and the amino acid requirement pattern.	Protein quality	DIAAS (%) = 100 × (mg of digestible dietary IAA in 1 g of the dietary test protein)/(mg of the same amino acid in 1 g of the reference protein)	The amount and profile of IAAs, including histidine (His), isoleucine (Ile), leucine (Leu), valine (Val), lysine (Lys), threonine (Thr), phenylalanine (Phe), methionine (Met), and tryptophan (Trp)	DIAAS score
Essential Amino Acid Index (EAAI)	Oser et al. [37]	1951	A geometric mean for calculating an overall comparison between all essential amino acids in the protein to be tested and all essential amino acids in the reference protein model.	Amino acid quality		All essential amino acids	EAAI
Index of Nutrition Quality (INQ)	Sorenson et al. [38]; Gholamalizadeh et al. [39]	1976, 2021	A method of quantitatively and qualitatively analyzing single foods, meals, and diets which has special significance for assessing clinical nutritional issues.	Nutrient amount	Equal to the amount of a nutrient in 1000 kcal of a food or diet divided by its RDA in 1000 kcal	Vitamin A, vitamin C, iron, vitamin D, vitamin E, thiamin, riboflavin, niacin, pantothenic acid, vitamin B6, folate, vitamin B12, copper, magnesium, zinc, calcium, and selenium	INQ scores
Naturally nutrient rich (NNR)	Drewnowski et al. [40]	2005	Mean percentage daily values (DVS) for 14 nutrients in 2000 kcal of food.	Nutrient density	NNR = ∑%DV2000 kcal/14	Protein, calcium, iron, vitamin A, vitamin C, thiamine, riboflavin, vitamin B12, folate, vitamin D, vitamin E, monounsaturated fat, potassium, and zinc	NNR score
Calories-for-nutrients (CFN)	Drewnowski et al. [40]	2005	The cost in calories required to obtain an additional 1% DV for a range of key nutrients.	Energy density	CFN = ED/(∑%DV100 g/13)	Protein, calcium, iron, vitamin A, vitamin C, thiamine, riboflavin, vitamin B6, vitamin B12, niacin, folic acid, magnesium, and zinc	CFN score
The ratio of recommended to restricted food score (RRR)	Scheidt et al. [41]	2004	The ratio of “good” to “bad” nutrients and to the energy content of the food, based on the food label.	Energy density	RRR = (∑%DVrecommended/6)/(∑%DVrestricted/5)	Six nutrients (protein, calcium, iron, vitamin A, vitamin C, and fiber) were defined a priori as desirable, whereas five nutrients (energy, saturated fat, cholesterol, sugar, and sodium) were defined as undesirable	RRR score
Dietary quality index (DQI)	Patterson et al. [42]	1994	An instrument to measure overall diet quality that reflects a risk gradient for major diet-related chronic diseases based on US dietary recommendations from Diet and Health.	Diet quality	The Diet and Health recommendations were weighted, cutoffs were developed for index scoring, and scores were summed across recommendations.	Energy from fat, energy from saturated fat, cholesterol, fruits and vegetables, grains and legumes, protein, sodium, and calcium	DQI summed score (0–16)
Dietary quality index–Revised (DQI-R)	Haines et al. [43]	1999	A revision of the DQI according to USDA data in 1994, reflecting the most current dietary guidance for the population.	Diet quality, variety	Each of the 10 components contributes 10 points to the total DQI-R score	Energy from fat, energy from saturated fat, dietary cholesterol (mg), recommended servings of fruit per day, recommended servings of vegetables per day, recommended servings of grains per day, calcium, RDA iron per day, dietary diversity, dietary moderation	Total DQI-R score (0–100)
Diet Quality Index–International (DQI-I)	Kim et al. [44]	2003	A composite measure of diet quality created to evaluate the healthiness of a diet not only within a country for monitoring purposes but also across countries for comparative work.	Dietary variety, adequacy, moderation, overall balance	Scores for each component are summarized in each of the four main categories, and the scores for all four categories are summed	Variety (overall variety and variety within protein sources), adequacy (fruits, vegetables, grains, fiber, protein, iron, calcium, vitamin C), moderation (total fat, saturated fat, cholesterol, sodium, empty calories foods), and overall balance (macronutrient ratio, fatty acid ratio)	Total DQI-I score (0–100)
Glycemic Index (GI)	Jenkins et al. [45]	1981	The glycemic effect of available carbohydrates in food relative to the effect of an equal amount of glucose.	Effect on human blood glucose	GI = (Iaucfood/Iaucglucose) × (Wt glucose/Wt available carbohydrate in food) × 100%	Available carbohydrate, the effect on human blood glucose	GI score (<55, low GI; 55–69, medium GI; >70 high GI)

**Table 2 nutrients-14-02352-t002:** Advantages and disadvantages of different types of methods.

Method Types		Main Approaches	Strengths	Weaknesses	Examples
**Based on food nutrient** **s**	Nutrient comparison	Nutrient detection; data analysis	Fast and simple	Unable to reflect the beneficial effects of foodstuffs	Comparison of nutrients in foodstuff with FCTs or control samples
	Chemical scoring	Nutrient detection; data analysis; calculation	Scientific and effective; further analysis of nutrient status	Individual scores can only reflect the nutritional characteristics of food from a certain angle; scores should be combined to reflect the nutritional properties of food	Amino acid score (AAS); protein-digestibility-corrected amino acid score (PDCAAS); digestible indispensable amino acid score (DIAAS); essential amino acid index (EAAI); index of nutrition quality (INQ); naturally nutrient rich (NNR); calories-for-nutrients (CFN)
**Based on sensory perceptions**		Description of color, aroma and taste	Intuitive; easily accepted by consumers	Needs professional researchers to complete; individual preferences vary widely	Sensory evaluation
**Based on the health effects and mechanisms of food**		Nutrient detection; calculation; cell experiments; animal experiments; human trials	Objective and fair; reflecting the effect of food/diet on human health, such as chronic disease prevention	Complicated experimental process; high experimental costs; long study cycles; accompanied by certain safety risks	Dietary quality index (DQI); dietary quality index–Revised (DQI-R); diet quality index–International (DQI-I); glycemic index (GI)

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
