# Peer review of "Evaluating the Nutritional Properties of Food: A Scoping Review"

_nutrients, 2022, doi:10.3390/nu14112352_

Round 1

Reviewer 1 Report

The review is well constructed and points out the attention of the difficult to accurately the comprehensive nutritional value of a food with a single indicator, and a systematic evaluation system is lacking. The purpose of this review was to establish an evaluation procedure for the nutritional properties of foodstuffs. The selection of a three-level evaluation method that covers the whole spectrum of a food’s nutritional characteristics was purposed. The method will be exploited by different scientific research.

Reviewer 2 Report

The authors have conveniently addressed all my comments and queries; therefore, I recommend to accept the revised version of the manuscript for publication at Nutrients.

Reviewer 3 Report

This paper seems to be two papers in one.

Paper 1 provides a review of food nutrition scoring/evaluation tools.

Paper 2 proposes a system for classifying food nutrition scoring tools. 

Each paper is interesting but each is incomplete.  Paper 1 reads like a

complete list of food nutrition scoring tools, but many of the tools are

old and it’s clear some commonly used tools (like the Healthy Eating

Index) are missing. I’m assuming the authors used to scoping review to

identify the tools. If so, perhaps the search criteria weren’t appropriate. 

There is also no clear presentation of the 16 articles ‘captured’ in the

reviewed. Paper 2.  The way that it’s presented, it seems that the system

for classifying food nutrition scoring/evaluation tools is an afterthought. 

If the authors think this system is important for the research community,

there should be more focus on it’s development and a more in depth

description of the Level’s. Finally in the Discussion, the “environmental

protection” issue is mentioned.  I think this should be address in section

3 when scoring tools are discussed.  I’m guessing there are already

scoring tools available for the sustainability of food items.

This manuscript is a resubmission of an earlier submission. The following is a list of the peer review reports and author responses from that submission.

Round 1

Reviewer 1 Report

Dear Authors,

The manuscript (nutrients-1662044) submitted for review is interesting. The manuscript is interesting but not well written, and I have a few comments.

Authors, Please note and address the following comments:

Abstract

I suggest that the authors reflect on the abstract. In this form, it is too general and, in my opinion, it is not interesting for the readers.

Material and methods - It is a pity that the authors did not write about the methods they were used to search for articles for review. In which databases were the articles searched? What was the key to choosing these references?

Summary

I don't see a clear authors' contribution to this article other than article collection. There is no in-depth analysis, possibly a SWOT analysis. It would be good to think and maybe just include in the table what the strength and weaknesses of these methods are and their uses. There is no guidance for the readers for uses this information. I don't know what significance this article has outside of a historical perspective.

Conclusions

What are the practical and theoretical implications of this review?

What are the authors' recommendations for scientists?

Technical Notes

  1. The citations in the text do not comply with the publisher's requirements.
  2. Figure 1, while interesting, is not clear and difficult to read.
  3. Figures 2 and 3 are illegible and are of low quality.
  4. The font in the References section does not match the MDPI publisher's requirements.

Reviewer

Reviewer 2 Report

Review of Manuscript ID: nutrients-1662044

Type of manuscript: Review

Title: Evaluation methods of food nutrition: a review of methods and highlights

Authors: Pei Wang, Jiazhang Huang, Junmao Sun, Rui Liu, Tong Jiang, Guiju Sun

General comment: the study reviews the general procedure for a correct evaluation of nutritional properties of foodstuffs. With the continuous development of new tools, techniques and research approaches in the field of nutrition, reviews such as the present one are the more and more necessary to update the completely new information. Indeed, this is not a comprehensive review of all methods described and used to test nutritional composition and potential bioactivity, but a systematized exposure of the general roadmap for nutritional evaluation of food. Perhaps the title stating that this is a review of methods (implying all methodology) should be changed and the meaning restricted. Some specific comments are detailed below:

Specific comments:

  • Pages should be correctly numbered. They seem to start again to page 1 after table 1.
  • Page 1, Last paragraph; regarding the sentence: single nutrient is far from enough to prevent the occurrence of chronic diseases, the authors should consider diseases provoked by vitamin or mineral (Fe, Ca) deficiencies, in which the lack of a single nutrient is enough to trigger the pathologic process.
  • Page 2, line 7; the authors should explain the meaning of wemedia.
  • Introduction, end of page 1 and beginning of page 2; perhaps the current concept of optimal nutrition, as substitutive of adequate nutrition, should be introduced and explained. I suggest a little more emphasis on the idea of nutrition not as a supply of nutrients but as an opportunity to prevent the onset of disease and maintain health. The concept is forestalled in these lines, but perhaps it should be exposed clearer.
  • Figure 1; although I am not in the field of food industry, the chart depicted in figure 1 including the classified catalogue of food production license revised by China State Administration for Market Regulation in 2020, contains some surprising subcategories such as liquor, pill, capsules…within the Health Food category. Perhaps a brief explanation of the classification of the chart would help to understand the meaning of the whole figure.
  • Page 4, epigraph 2.1.2; biological value should not be considered a method, but rather a calculation or evaluation of the proportion of protein retained in the body for growth and/or maintenance and expressed in percent of nitrogen absorbed.
  • Page 1 AFTER TABLE 1 (see first comment regarding page numbering); the concept that “healthy” is always associated with “tasty” is highly contradictory considering the RRR index depicted in table 1, where nutrients such as sugar and sodium, which add taste to foodstuff, are defined as undesirable.
  • Figure 2; level 2, text within the lowest speech bubble, it should say benefit of dietary patterns. Level 3, text within the upper speech bubble the sentence says ents on gene structure…The text should be checked.
  • Figure 3; perhaps figure 3 should be introduced and commented before figure 2, since figure 3 shows the food nutrition evaluation procedure and figure 2 shows a step further, the three levels of evaluation depicted at the end of figure 3.
  • Discussion, line 5; clarified clearly sound redundant.
